# Using Machine Learning to Identify Feelings of Energy and Fatigue in Single-Task Walking Gait: An Exploratory Study

Ahmed M. Kadry [1,2], Ahmed Torad [1,2], Moustafa Ali Elwan [1,3], Rumit Singh Kakar [4], Dylan Bradley [5], Shafique Chaudhry [6,7] and Ali Boolani [1,8,*]

1   Department of Physical Therapy, Clarkson University, Potsdam, NY 13699, USA; ahmed_tabia@pt.kfs.edu.eg (A.M.K.); ahmed_alimohamed@pt.kfs.edu.eg (A.T.); mostafa.ali@pt.bsu.edu.eg (M.A.E.)
2   Faculty of Physical Therapy, Kafrelsheik University, Kafr El Sheik 33516, Egypt
3   Faculty of Physical Therapy, Beni-Suef University, Beni-Suef 62521, Egypt
4   Human Movement Science Department, School of Health Sciences, Oakland University, Rochester, MI 48309, USA; kakar@oakland.edu
5   Canino School of Engineering Technology, State University of New York at Canton, Canton, NY 13617, USA; bradley.dylan028@gmail.com
6   David D. Reh School of Business, Clarkson University, Potsdam, NY 13699, USA; schaudhr@clarkson.edu
7   Department of Computer Science, Clarkson University, Potsdam, NY 13699, USA
8   Department of Biology, Clarkson University, Potsdam, NY 13699, USA
*   Correspondence: aboolani@clarkson.edu

**Abstract:** The objective of this study was to use machine learning to identify feelings of energy and fatigue using single-task walking gait. Participants ($n = 126$) were recruited from a university community and completed a single protocol where current feelings of energy and fatigue were measured using the Profile of Moods Survey–Short Form approximately 2 min prior to participants completing a two-minute walk around a 6 m track wearing APDM mobility monitors. Gait parameters for upper and lower extremity, neck, lumbar and trunk movement were collected. Gradient boosting classifiers were the most accurate classifiers for both feelings of energy (74.3%) and fatigue (74.2%) and Random Forest Regressors were the most accurate regressors for both energy (0.005) and fatigue (0.007). ANCOVA analyses of gait parameters comparing individuals who were high or low energy or fatigue suggest that individuals who are low energy have significantly greater errors in walking gait compared to those who are high energy. Individuals who are high fatigue have more symmetrical gait patterns and have trouble turning when compared to their low fatigue counterparts. Furthermore, these findings support the need to assess energy and fatigue as two distinct unipolar moods as the signals used by the algorithms were unique to each mood.

**Keywords:** energy; fatigue; falls; machine learning; walking gait

## 1. Introduction

Falls are a major health concern for adults as they age. The Center for Disease Control (CDC) recently reported that more than 33% of adults over the age of 65 years of age suffered a fall at least once a year, with many suffering multiple falls [1]. Approximately 10% of falls result in serious injuries, and annual fall related healthcare costs are estimated at $50 billion [2,3]. Research suggests that there are numerous factors associated with falls, with the most significant risk being related to impaired balance and gait [4]. While there are many reasons for balance and gait impairment, a recent area of interests for gait and balance researchers has been the role of fatigue in functional balance and gait declines in older adults. For example, Grobe and colleagues postulated that mental fatigue would lead to decreased balance control and changes in gait parameters in older adults, thus increasing fall risks [5]. However, studies examining objective and subjective measures of fatigue and their impact on gait and balance in older adults have reported mixed results [6–10].

Significant evidence exists on the use of machine learning to identify fatigue through the use of inertial movement unit (IMUs) sensors [11–16]. However, the primary limitation of these studies [12–16] is that fatigue was measured as a perception of effort through using a Ratings of Perceived Exertion (RPE) scale. While many researchers use RPE as a measure of fatigue, exercise studies suggest that this scale may not be a good measure of subjective perceptions of fatigue [17–21].

Another limitation in current literature seeking to identifying fatigue through human movement is that these studies measured fatigue as the lack of energy [6–10], rather than measuring energy and fatigue as two separate unipolar moods. Significant evidence exists that energy and fatigue are distinct unipolar moods with their own unique biological [22–27] and behavioral correlates [24,28,29]. Recently, studies that have examined energy and fatigue as two separate unipolar moods have also reported their distinct impact on gait and balance [30–35]. For example, Boolani and colleagues [30] examined feelings of energy and fatigue as two separate unipolar moods, and the results of that study reported that declines in feelings of energy resulted in declines in Berg Balance Test (BBT) scores in a group of older adults (>65 years). Other findings suggest that feelings of energy are associated with performance on gait associated functional tests [33,34], while feelings of fatigue were not. However, the gait and balance parameters reported in these studies [30,33,34] were gross measures that provided limited information for researchers interested in identifying feelings of fatigue or energy through human movement.

Studies that have taken a more granular approach to identifying the influence of feelings of energy and fatigue as two distinct unipolar moods on gait and balance have primarily examined these two constructs in young adults. These studies provide evidence that the two moods influence gait and balance in very distinct ways [31,32,35]. For example, Kowalski and colleagues [32] report feelings of energy being associated with upper extremity and lumbar movement variability, while feelings of fatigue were associated with lateral step variability and step length [31]. Mahoney and colleagues [35] suggest that current feelings of energy were associated with gait patterns most often associated with increased risks of tripping, while current feelings of fatigue were associated with slowing of gait, most often associated with increased guarding. Taken together, these studies suggest that researchers interested in identifying current feelings of energy and fatigue using machine learning (ML) should examine single-task walking gait.

Due to limitations in the current literature in identifying feelings of energy and fatigue as two distinct moods using machine learning, it is advisable to perform an exploratory study in healthy young populations that do not report as many health issues that may influence gait [36]. Therefore, to provide researchers with interesting findings without accounting for various health conditions that may influence gait in older adults, this study chose to use young adults to guide future targeted studies in an older adult population.

The objectives of this current study were to use machine learning to identify: (i) feelings of energy; (ii) feelings of fatigue; (iii) individuals who are high energy/low fatigue, high energy/high fatigue, low energy/high fatigue and low energy/low fatigue; using walking gait. Each of the three objectives of the study have the following sub-objectives: (a) Identify the most important features of gait needed to identify these moods; (b) Identify the most accurate models; (c) Identify gait characteristics that differ between groups.

## 2. Methods

### 2.1. Study Design

A caffeine- and exercise-controlled, cross-sectional design was used to collect data between 1 June 2018 and 1 May 2019. Participants were split into groups based on self-reported feelings of energy and fatigue and were grouped into either high energy or low energy; and high fatigue or low fatigue: high energy and low fatigue, high energy and high fatigue, low energy and low fatigue and low energy and high fatigue.

### 2.2. Participants

Participants were recruited from the community through word of mouth, flyers, recruitment at the university through announcement in large classes (>30 students), and through campus-wide emails. Inclusion criteria for the study were as follows: between the ages of 18 and 36, ability to ambulate without an assistive device for >2 min and stand independently without pain and/or discomfort. Exclusion criteria for this study were as follows: diagnosis of a neurological condition (i.e., stroke, Parkinson's disease), lower extremity orthopedic surgery within the last 6 months, wounds or abscesses on the plantar surface of the feet, and un-correctable visual impairments. A total of 144 volunteers were recruited for this study and 126 qualified to participate. Participants were eliminated due to age ($n$ = 14), presence of neurological condition ($n$ = 1), un-correctable visual impairment ($n$ = 1), lower extremity orthopedic surgery within the last 6 months ($n$ = 2). All participants were informed of the procedures prior to participation and signed an informed consent form. The experimental procedures conformed to the Declaration of Helsinki and were approved by Clarkson University Institutional Review Board (IRB) approval (approval #18.39.1).

### 2.3. Instruments

2.3.1. Self-Reported Feelings of Energy and Fatigue

The 30-item Profile of Mood States–Short Form (POMS-SF) was used to measure feelings of energy (vigor) and fatigue as two separate unipolar moods [37]. The POMS-SF measure is used to asses 6 different mood states. However, for the purposes of this study only feelings of energy (vigor) and fatigue were calculated and used. Participants were asked to indicate their current intensity of subjective mood states on a 5-point scale ranging from 0 (Not at all) to 4 (Extremely). Both of the mood states were scored on the basis of a sum of five questions: fatigue = worn out + fatigued + exhausted + sluggish + weary; energy (vigor) = lively + active + energetic + full of pep + vigorous. Among healthy participants, the Cronbach's α, a measure of internal consistency, has been reported as to be 0.93 for both energy (vigor) and fatigue [38]. The values of Cronbach's α for this study were 0.810 (energy) and 0.817 (fatigue).

2.3.2. Gait

Gait was assessed using the APDM mobility Lab™ (APDM Inc., Portland, OR, USA) during a 2-min walk around a 6 × 1-m oval track, a protocol most often used in literature when machine learning is used to identify mood states using gait parameters [39,40]. The APDM mobility Lab™ system consists of a set of wireless, body-worn Opal™ inertial sensors, each with a docking station, an access point for wireless data transmission and sub-millisecond synchronization of the independent sensors. The Opal™ inertial sensors contain a tri-axial accelerometer, a trial axial gyroscope, and a tri-axial magnetometer and have a range of 6 m from the docking station. The accelerometers measure linear acceleration, gyroscopes measure angular velocity, and the magnetometers measure heading with respect to the earth's magnetic field. For this study, seven Opals™ were attached to the body using Velcro™ straps: lumbar region (5th lumbar vertebra), sternum (body of sternum superior to the xyphoid process), forehead (middle of the frontal bone, approximately 2.5 cm above the nasal bone), right and left foot (on the metatarsals, directly superior to the metatarsophalangeal joint), and the right and left wrist (immediately superior to the radio-ulnar joint). Participants were asked to walk to a cone that was placed 6 m from the starting point, make a 180° left turn around the cone and then come back to the starting cone. The cones were placed 6 m apart and participants were asked to walk back and forth for 2-min at whatever speed they felt most comfortable. This method of gait assessment using the APDM Opal™ monitors has been validated and proven reliable [41–43], and is the most commonly used gait assessment method for identifying subjective parameters using walking gait [39,40,44].

### 2.4. Procedure

After screening for inclusion and exclusion criteria, participants were schedule for one testing session lasting approximately 75 min. Participants were asked to refrain from consuming alcohol, caffeine, medications, or illicit drugs for at least 24 h prior to testing. When participants arrived in lab, they completed a pre-test survey using SurveyMonkey Inc. (San Mateo, CA, USA) using a Hewlett Packard Pavilion 15.6" Flagship Laptop (model #B018YIGHVK, Hewlett Packard, Houston, TX, USA), to determine whether they had followed pre-testing instructions. If participants had followed pre-testing instructions, they were assigned a random 5-digit ID using randomizer.org, fitted for the APDM mobility monitors, and completed a series of surveys asking them about their activity, diet, and sleep over the last 24 h. Participants then completed the POMS-SF to assess their current mood states. After completion of the surveys, participants were asked to complete a 2-min modified Clinical Test of Sensory Interaction in Balance (mCTSIB) protocol to measure balance (data not used in this study) and 2-min walk. After completion of the walk, participant's height was measured using a stadiometer (SECA model 220 Crothal Healthcare, Chino, CA, USA) and their weight was measured using the Tanita Body Composition Analyzer TBF-410 (TBF-410, Tanita Corporation, Tokyo, Japan).

### 2.5. Statistical Analysis

All preliminary and primary analyses were conducted using Python (version 3.8.5, Python Software Foundation, Wilmington, DE, USA).

#### 2.5.1. Pre-Processing of Data

Data from SurveyMonkey.com were downloaded and exported into Microsoft Excel (Microsoft Inc., Redmond, WA, USA), where mood data were scored. Gait and postural data were exported as h5 files from the APDM software into Python where means and variation were calculated for each gait parameter. For upper and lower extremity variables mean and standard deviation for each limb were calculated and mean measure for each extremity limb variables (i.e., mean gait speed) and imbalance between each extremity limb (i.e., variation in gait speed between legs) were also calculated. The following formulas were used:

Mean for each gait parameter (i.e., turn velocity)

$$\left( mean\ turn\ velocity = \frac{sum\ of\ velocity\ of\ all\ turns}{number\ of\ turns} \right)$$

Variation for each gait parameter (i.e., standard deviation in turn velocity)

$$\left( variation\ in\ turn\ velocity = \sqrt{\frac{\sum_{i=1}^{i=number\ of\ turns}(turn\ velocity - mean\ turn\ velocity)^2}{Number\ of\ turns}} \right)$$

Mean gait parameter for individual limb (i.e., mean gait speed for right leg only)

$$\left( mean\ gait\ speed\ right\ leg = \frac{sum\ of\ gait\ speed\ of\ right\ leg\ during\ 2\text{-}minute\ walk}{number\ of\ steps\ taken\ by\ right\ leg\ in\ 2\text{-}minute\ walk} \right)$$

Variation in movement for each individual limb (i.e., standard deviation for right leg gait speed only/intra-limb variation in movement)

$$\left( variation\ in\ right\ leg\ gait\ speed = \sqrt{\frac{\sum_{i=1}^{i=number\ of\ steps\ on\ right\ leg}(right\ leg\ gait\ speed - mean\ right\ leg\ gait\ speed)^2}{Number\ of\ steps\ on\ right\ leg}} \right.$$

Mean gait parameter for both limbs of upper and lower extremity (i.e., mean gait speed)

$$Mean\ gait\ speed = \frac{\sum(gait\ speed\ right\ leg + gait\ speed\ left\ leg)}{Total\ number\ of\ steps}$$

Imbalance in gait parameters between limbs (i.e., variation in gait speed between limbs/inter-limb variation in movement) to assess for consistency of movement and synchronization of movement on both sides [45,46].

$$Imbalance\ in\ gait\ speed\ between\ limb = \sum \frac{gait\ speed\ right\ leg - gait\ speed\ left\ leg}{gait\ speed\ right\ leg + gait\ speed\ left\ leg} / Total\ number\ of\ steps$$

After calculation of gait parameters, data of all gait variables were merged with the calculated mood data.

Data were then further pre-processed for training. There were 314 missing values distributed across 23 variables (total 3312 values). To fill in missing values, Nearest Neighbor Imputer (NNImputer) with a 5 neighbor imputer was used to replace missing values [47]. The NNImputer, 5 neighbor imputer techniques involve taking the mean of the neighboring 5 values on both sides of the data and replacing the missing value with that mean [47]. Data were then screened for outliers and winsorized to mitigate the influence of extreme values [48]. Winsorization involves taking extreme values and replacing them with the value that corresponds with a certain percentile of the original distribution. All gait variables in this study were rescaled to recode values past the 95th percentile to the value of the 95th percentile [48].

2.5.2. Primary Analyses

- Classification of participants

Participants were classified based on the median scores of both energy (vigor) and fatigue scores on the POM-SF. Chi-squares were used to identify differences in sex distribution between groups and student T-tests were used to identify differences in age, height, weight and Body Mass Index (BMI) between each group (2 groups) An ANOVA was used to identify differences for age, height, weight and BMI between the groups (4 groups) that were split by their energy and fatigue status.

*Objective 1:*

Participants who reported a score > 5 (median value) on the energy (vigor) portion of the POMS were classified as "high energy", while individuals who scores ≤ 5 were classified as "low energy".

*Objective 2:*

For fatigue, participants who reported a score > 3 (median value) on the fatigue portion of the POMS-SF were classified as "high fatigue", while participants who reported scores ≤ 3 were classified as "low fatigue".

*Objective 3:*

Using the sub-categories from Objectives 1 and 2, participants were classified into the following groups: high energy and high fatigue, high energy and low fatigue, low energy and high fatigue, and low energy and low fatigue.

For each objective, the following analyses were conducted:

*Sub-Objective 1: Important Feature Selection*

When recording high-dimensional features, not every feature is equally important, and there may be many redundant features that are of less importance. Therefore, to optimize the number of features for each classification model, a Recursive Feature Elimination with cross-validation model using Random Forest Classifier as an estimator were used. Furthermore, selection from model or forward Sequential Feature Selector were used to extract the dataset with the 12 most important features to optimize accuracy of the models [49].

*Sub-Objective 2: Model Training and Evaluation*

After sorting the features, the dataset was used to train the model through both Classifiers and Regressors respectively. For the Classifiers, the categories for energy, fatigue and energy + fatigue were used to train each of the models. Models used for classification were Logistic Regression, Decision Tree Classifier, Random Forest Classifier, Gradient Boosting Classifier, K Nearest Neighbour (KNN) Classifier, Multi-Layer Perceptron (MLP) Classifier, Support Vector Classifier (SVC) Model, Gaussian Naïve Bayes Model, and Bagging Classifier.

Both moods were treated as continuous variables for the Regressor models. Models used for regressors were: Decision Tree Regressor, Random Forest Regressor, Ridge Regression, Lasso Regression, Stochastic Gradient Descent (SGD) Regression MLP Regressor, Support Vector Regressor (SVR) Model, Gradient Boosting Regressor (GBR) Model, Neighbors Regressor, and Elastic Net Model.

Each model was trained in a 10-fold cross-validation manner in order to avoid problems such as overfitting or selection bias to some degree [50]. Regressor models were also assessed using bootstrapping, as this allows validation of models with small sample sizes [51]. The models were randomly split into the training set (90%) and the test set (10%). Classification models were evaluated based on mean and median "Accuracy" and the area under the curve (AUC) of Receiver Characteristic Operator (ROC), while Mean Absolute Error (MAE) was used to assess Regressor accuracy. Correlation coefficients ($R^2$) were also assessed between the predicted mood score and the self-reported mood score on the regressor models [44].

*Sub-Objective 3: Differences in gait characteristics between groups*

To identify differences between groups (low vs. high energy; low vs. high fatigue; high energy/high fatigue vs. high energy/low fatigue vs. low energy/high fatigue vs. low energy/low fatigue), Multivariate Analysis of Covariance (MANCOVA) were used with sex, age, weight, and height used as co-variates. Prior to the primary analysis, all gait variables were screened between groups for normality assumptions using a combination of histograms and the Shapiro-Wilks test for normality. For non-normally distributed variables, exponential, power, arscine and logarithmic transformations were applied; however, the histograms and Shapiro–Wilks tests did not differ much from the original variables. Therefore, using a combination of large sample theory [52,53] and the fact that there are no non-parametric versions of a MANCOVA available, a MANCOVA was used. A full-factorial general linear model, with a polynomial multivariate contrast was used to assess differences between the different independent groups. A Bonferroni adjusted post hoc pair-wise comparison using estimated marginal mean was used where necessary and $\alpha$ levels were set at 0.05. Post hoc Cohen's *d* was calculated using adjusted means and pooled standard deviations. All analyses and *p*-values presented in the results represent corrected values.

## 3. Results

### 3.1. Feelings of Energy (Vigor)

There were 63 (males = 20, females = 43) participants classified as low energy and 63 (males =27, females = 36) as high energy. There were no significant differences between sex, age, height, weight, and BMI between groups ($p > 0.05$). See Table 1.

#### 3.1.1. Feature Importance

The top 12 features selected for the most accurate classifier model were the variation in turn velocity, imbalance in double limb support time between legs, variation in lumbar flexion/extension range of motion (ROM), variation in right leg stride length, variation in left leg circumduction, mean right leg double leg support time, mean maximum lumbar rotation to the left, imbalance in cadence between legs, weight, variation in right leg cadence, anticipatory postural adjustment (APA) first step time and mean neck rotation range of motion to the right. Features and level of importance are reported in Table 2, while all features and their levels of importance are reported in Supplementary Table S1.

**Table 1.** Participant characteristics.

| Variable | Energy | | | Fatigue | | | Energy + Fatigue | | | | |
|---|---|---|---|---|---|---|---|---|---|---|---|
| | Low Energy | High Energy | Sig. | Low Fatigue | High Fatigue | Sig. | High Energy/High Fatigue | Low Energy/High Fatigue | High Energy/Low Fatigue | Low Energy/Low Fatigue | Sig. |
| Male:Female | 20:43 | 27:36 | 0.197 | 24:29 | 23:50 | 0.114 | 13:20 | 14:16 | 10:30 | 10:13 | 0.246 |
| Age | 24.25 ± 3.94 | 23.6 ± 3.52 | 0.330 | 23.87 ± 3.93 | 23.97 ± 3.62 | 0.877 | 23.03 ± 2.99 | 24.23 ± 3.97 | 24.75 ± 3.93 | 23.39 ± 3.9 | 0.817 |
| Height | 172.56 ± 8.42 | 173.61 ± 8.61 | 0.491 | 173.77 ± 6.84 | 172.58 ± 9.54 | 0.439 | 173.49 ± 9.3 | 173.74 ± 7.94 | 171.83 ± 9.79 | 173.82 ± 5.24 | 0.805 |
| Weight | 71.65 ± 12.86 | 74.46 ± 13.66 | 0.236 | 74.08 ± 12.66 | 72.31 ± 13.77 | 0.464 | 73.45 ± 14.12 | 75.58 ± 13.28 | 71.38 ± 13.57 | 72.12 ± 11.81 | 0.859 |
| BMI | 23.88 ± 3.68 | 24.63 ± 3.81 | 0.263 | 24.36 ± 3.53 | 24.18 ± 3.92 | 0.791 | 24.39 ± 4.22 | 24.89 ± 3.36 | 24 ± 3.7 | 23.66 ± 3.7 | 0.643 |

**Table 2.** Top 12 most important features for best classifier models.

| Level of Importance | Energy * | | Fatigue * | | Energy + Fatigue ** | |
|---|---|---|---|---|---|---|
| Ranking | Relative Importance | Gait Characteristic | Relative Importance | Gait Characteristic | Relative Importance | Gait Characteristic |
| 1 | 0.159 | Variation in Turns Velocity (°/s) | 0.146 | Mean Max. Lumbar R Lat. bend (°) | 0.138 | Variation Max. Lumbar Rot ROM to the right (°) |
| 2 | 0.154 | Imbalance in DLS-GCT between legs (%) | 0.142 | Variation L Heel Strike Angle (°) | 0.108 | Variation in Turns Angle (°) |
| 3 | 0.124 | Variation Lumbar Flexion/Extension ROM (°) | 0.139 | Variation in Turns Angle (°) | 0.046 | APA 1st step ROM (°) |
| 4 | 0.094 | Variation R leg Stride Length (m) | 0.131 | Variation in R leg elevation at mid-stance (cm) | 0.031 | Variation L DLS-GCT (%) |
| 5 | 0.091 | Variation L leg Circumduction (cm) | 0.087 | Mean L Toe Out Angle (°) | −0.031 | Variation L Stride Length (m) |
| 6 | 0.086 | Mean R leg DLS-GCT (%) | 0.08 | Mean Min. Lumbar Flexion/Extension ROM (°) | −0.092 | Mean Max. Neck Flexion/Extension ROM (°) |
| 7 | 0.067 | Mean Max. Lumbar Rot ROM (°) | 0.067 | Variation Max. Lumbar L Rot ROM (°) | −0.092 | Mean Neck Flexion/Extension ROM (°) |
| 8 | 0.049 | Imbalance in Cadence between legs (%) | 0.060 | Imbalance in step variability between legs (%) | −0.092 | Variation L leg Circumduction (cm) |
| 9 | 0.047 | Weight (kg) | 0.057 | Variation R Toe Off Angle (°) | −0.108 | Lateral APA Peak (m/s$^2$) |
| 10 | 0.046 | Variation R leg cadence (s) | 0.043 | Mean Trunk Flexion/Extension ROM (°) | −0.108 | Variation R leg elevation at mid-stance (cm) |
| 11 | 0.042 | APA 1st step time (s) | 0.033 | Variation Max. Lumbar R Rot ROM (°) | −0.108 | Variation Step Time (s) |
| 12 | 0.042 | Mean Neck R Rot ROM (°) | 0.016 | APA 1st step Time (s) | −0.123 | APA 1st step Time (s) |

* Importance calculated by Gini importance; ** Importance calculated by permutation importance; ° = degrees; APA = Anticipatory Postural Adjustment; DLS-GCT = Double Support as a percentage of Gait Cycle Time; L = left; Lat. = Lateral; Max. = Maximum; Min. = minimum; R = right; ROM = range of motion.

For the most accurate regressor model, the number of turns, mean maximum neck bending to the right side, mean minimum neck flexion and extension ROM, mean arm swing velocity of both arms, imbalance in cadence between legs, mean minimum right leg cadence, mean maximum lumbar flexion/extension ROM, mean gait speed of both legs, mean cadence of both legs, mean minimum cadence on left leg, and imbalance in gait cycle time between legs were the most important variables. Feature importance is reported in Table 3, while feature importance for all features and their levels of importance are reported in Supplementary Table S1.

**Table 3.** Top 12 most important features for the best regressor models.

| Level of Importance | Energy * | | Fatigue * | |
|---|---|---|---|---|
| Ranking | Relative Importance | Gait Characteristic | Relative Importance | Gait Characteristic |
| 1 | 0.162 | Number of turns (#) | 0.230 | APA 1st step ROM (°) |
| 2 | 0.140 | Mean Max. Neck R Lat. bend ROM (°) | 0.228 | Mean Max. Neck Flex/Ext ROM (°) |
| 3 | 0.103 | Mean Min. Neck Flex/Ext ROM (°) | 0.085 | Imbalance Mean Toe Out Angle between legs (°) |
| 4 | 0.089 | Mean arm swing velocity (°/s) | 0.084 | Mean Max. Neck L Lat. bend ROM (°) |
| 5 | 0.086 | Imbalance in cadence between legs (%) | 0.067 | Variation R DLS-GCT (%) |
| 6 | 0.080 | Mean R Cadence (steps/s) | 0.066 | Variation R Arm Swing Velocity (°) |
| 7 | 0.073 | Mean Max. Back Flex/Ext ROM (°) | 0.061 | Variation L Cadence (steps/s) |
| 8 | 0.073 | Mean Gait Speed of both legs (m/s) | 0.053 | Mean Neck R Lat. bend ROM (°) |
| 9 | 0.063 | Mean Cadence of both legs (steps/s) | 0.050 | Variation Max. Lumbar R Rot ROM (°) |
| 10 | 0.056 | Mean L Cadence (steps/s) | 0.045 | Variation R SLS-GCT (%) |
| 11 | 0.044 | Imbalance in Gait Cycle Time between legs (%) | 0.021 | Imbalance in Gait Cycle Time between legs (%) |
| 12 | 0.031 | Sex | 0.009 | Sex |

* Sequential Feature Importance; # = number; ° = degrees; APA = Anticipatory Postural Adjustment; DLS-GCT = Double Support as percent of gait cycle time; Flex/Ext = flexion/extension; L = left; Lat. = Lateral; Max. = Maximum; Min. = minimum; R = right; ROM = range of motion; SLS-GCT = Single Limb Support as percentage of gait cycle time.

### 3.1.2. Model Training

The most accurate classification model with the highest AUC ROC for identifying individuals with low and high energy was the Gradient Boosting Classifier model. The mean accuracy of the model was 74.34% (95% CI 0.708–0.779), the median accuracy was 75%, and the AUC ROC was 0.806. The highest accuracy for all models was 100% (0.3% of all models run). Regressor models are presented both with K-fold cross-validation and the bootstrapped method. The best regressor model for both with K-fold cross-validation and bootstrapped method was the Random Forest Regressor, with a mean MAE of 0.005 for the K-fold model and 0.006 for the bootstrapped model. Mean $R^2$ for the cross-validated models was 0.310, while the bootstrapped model had a mean $R^2$ of 0.884, with a maximum $R^2$ of 0.900 (Table 4).

**Table 4.** Best classifier and regressor models.

| Construct | Classifier | Mean | 95% CI | Minimum | Q1 | Q2 | Q3 | Maximum |
|---|---|---|---|---|---|---|---|---|
| Energy | Gradient Boosting Classifier | 0.743 | 0.708–0.779 | 0.539 | 0.673 | 0.750 | 0.769 | 1 |
| Fatigue | Gradient Boosting Classifier | 0.742 | 0.696–0.788 | 0.461 | 0.692 | 0.760 | 0.846 | 0.923 |
| Energy/Fatigue | Gaussian Naïve Bayes | 0.455 | 0.398–0.512 | 0.166 | 0.314 | 0.461 | 0.538 | 0.833 |
| **Regressor R$^2$** | | | | | | | | |
| Energy | Random Forest Regressor—Bootstrapped | 0.884 | 0.86–00.90 | 0.863 | 0.875 | 0.881 | 0.888 | 0.900 |
| Energy | Random Forest Regressor—with K-fold | 0.310 | 0.30–0.32 | 0.30 | 0.281 | 0.283 | 0.321 | 0.321 |
| Fatigue | Random Forest Regressor—Bootstrapped | 0.886 | 0.859–0.91 | 0.862 | 0.876 | 0.885 | 0.895 | 0.895 |
| Fatigue | Random Forest Regressor—with K-fold | 0.349 | 0.2–0.498 | 0.239 | 0.308 | 0.340 | 0.397 | 0.397 |
| **Regressor Mean Absolute Error** | | | | | | | | |
| Energy | Random Forest Regressor—Bootstrapped | 0.006 | 0.004–0.008 | 0.004 | 0.005 | 0.006 | 0.007 | 0.008 |
| Energy | Random Forest Regressor—with K-fold | 0.005 | 0.003–0.006 | 0.005 | 0.005 | 0.005 | 0.004 | 0.004 |
| Fatigue | Random Forest Regressor—Bootstrapped | 0.005 | 0.003–006 | 0.004 | 0.004 | 0.005 | 0.005 | 0.006 |
| Fatigue | Random Forest Regressor—with K-fold | 0.007 | 0.005–009 | 0.004 | 0.006 | 0.007 | 0.008 | 0.001 |

### 3.1.3. Differences in Gait Characteristics between Groups

Only 2 the top 12 features for the classifier models were significantly different between groups. All significant findings are reported in Table 5, and all variables examined are reported in Supplementary Table S1.

Individuals who reported being more energetic had higher variation in right leg cadence, stride length, double leg support time as a percentage of gait cycle time. More energetic individuals also had less imbalance between legs for double leg support time, higher variations in both left and right leg gait speed and higher variations in left leg toe off angle. Furthermore, when examining upper extremity movement, individuals who reported being energetic had faster right arm swing velocity, faster average arm swing velocity, greater range of motion (ROM) in both the left and right arm, and overall greater but less varied ROM for both arms. There were no significant differences between high- and low-energy individuals for gait initiation, neck, lumbar or trunk movement, as well as no difference in how they turn.

**Table 5.** Differences between high- and low-energy groups (mean ± SD).

| | Variable | Low Energy | High Energy | Sig. | *d* |
|---|---|---|---|---|---|
| Leg | Variation R cadence (steps/min) | 2.36 ± 0.76 | 2.69 ± 0.87 | 0.024 | −0.40 |
| | Variation R DLS-GCT (%) | 1.08 ± 0.23 | 1.18 ± 0.29 | 0.040 | −0.38 |
| | Imbalance in DLS-GCT between legs (%) | 0.71 ± 0.42 | 0.52 ± 0.43 | 0.047 | 0.37 |
| | Variation L gait speed (m/s) | 0.046 ± 0.015 | 0.052 ± 0.019 | 0.041 | −0.41 |
| | Variation R gait speed (m/s) | 0.047 ± 0.013 | 0.054 ± 0.017 | 0.009 | −0.47 |
| | Variation L toe off angle (°) | 1.34 ± 0.44 | 1.54 ± 0.5 | 0.019 | −0.45 |
| | Variation R stride length (m) | 0.038 ± 0.01 | 0.042 ± 0.012 | 0.044 | −0.36 |
| Arm | Mean R arm swing velocity (°/s) | 162.54 ± 53.08 | 183.91 ± 60.57 | 0.018 | −0.40 |
| | Mean swing velocity for both arms (°/s) | 177.57 ± 51.69 | 198.41 ± 61.14 | 0.017 | −0.40 |
| | Mean L ROM (°) | 42.85 ± 16.31 | 48.43 ± 17.78 | 0.033 | −0.38 |
| | Mean R ROM (°) | 36.05 ± 13.85 | 42.07 ± 15.81 | 0.017 | −0.43 |
| | Mean ROM for both arms (°) | 39.66 ± 14.04 | 45.08 ± 15.26 | 0.020 | −0.41 |
| | Imbalance in arm ROM between arms (%) | 14.6 ± 10.04 | 11.6 ± 8.49 | 0.038 | 0.38 |

All *p*-values reported are Bonferroni adjusted values: ° = degrees; DLS-GCT = double support as a percentage of gait cycle time; L = left; R = right side; ROM = range of motion.

### *3.2. Feelings of Fatigue*

There were 63 (males = 24, females = 29) in the low-fatigue group and 63 in the high-fatigue group (males = 23, females = 50). There was no significant difference in sex, age, height, weight, and BMI between groups (Table 1).

#### 3.2.1. Feature Importance

The top 12 most important features for the best classifier model were mean of the maximum lateral bend of the lumbar spine, variation in left leg heel strike angle, variation in turn angles, variation in right leg elevation during mid-swing, mean left leg toe out angle, mean minimum lumbar flexion/extension ROM, variation in maximum lumbar rotation on the left side, imbalance in lateral step-variability between legs, variation in right leg toe off angle, mean trunk flexion/extension ROM, variation in maximum lumbar rotation ROM on the right side, and anticipatory postural adjustment first step time (see Table 2 for relative importance and Supplementary Table S1 for feature importance for all variables).

For the best regressor models, the top 12 most important features include range of motion for first step in during gait initiation, mean maximum neck flexion/extension, imbalance in mean toe out angle between legs, mean maximum neck bending on the left side, variation in right leg double leg support time, variation in right arm swing velocity, variation in left leg cadence, mean lateral neck bending to the right side, variation in maximum lumbar rotation to the right side, variation in right leg double leg support time, and imbalance in gait cycle time between legs. See Table 3 for relative importance and Supplementary Table S1 for feature importance for all variables.

#### 3.2.2. Model Training

The most accurate classification model with the highest AUC ROC was the Gradient Boosting Classifier model, with 74.23% accuracy (95% CI 0.696–0.788), with a median accuracy of 75.96% and an AUC ROC of 0.819. The most accurate classification model for fatigue had a 92.31% accuracy rate. Random Forest Regressor was the most accurate regressor model. The mean MAE for the K-fold model is 0.007, and the mean MAE for

the bootstrapped model is 0.005. The mean $R^2$ for the K-fold model is 0.349, while for the bootstrapped it is 0.884, with a maximum $R^2$ of 0.895 (Table 4).

### 3.2.3. Differences in Gait Characteristics between Groups

Only 4 of the top 12 features in the model were significant different between groups. All significant findings are reported in Table 6, and Supplementary Table S1 contains the results of all the analyses.

**Table 6.** Differences between high and low fatigue groups.

|  | Variable | High Fatigue | Low Fatigue | Sig. | *d* |
|---|---|---|---|---|---|
| Gait Initiation | APA first step time (s) | $0.59 \pm 0.06$ | $0.56 \pm 0.07$ | 0.032 | 0.39 |
| Lumbar | Mean Min. Lumbar Flex/Ext ROM (°) | $-1.68 \pm 4.81$ | $-3.16 \pm 3.9$ | 0.043 | 0.37 |
|  | Variation Max. Lumbar R Rot ROM (°) | $4.31 \pm 2.97$ | $5.36 \pm 3.38$ | 0.024 | $-0.40$ |
|  | Variation Max. Lumbar L Rot ROM (°) | $4.44 \pm 2.92$ | $5.48 \pm 3.4$ | 0.024 | $-0.40$ |
| Trunk | Variation Trunk Flex/Ext ROM (°) | $1.1 \pm 0.35$ | $1.2 \pm 0.39$ | 0.035 | $-0.38$ |
| Leg | Imbalance in Gait Speed between legs (%) | $0.89 \pm 0.67$ | $1.15 \pm 0.73$ | 0.019 | $-0.43$ |
|  | Mean L Toe Off Angle (°) | $37.21 \pm 2.91$ | $35.93 \pm 3.41$ | 0.041 | 0.37 |
|  | Mean R Toe Off Angle (°) | $37.32 \pm 2.78$ | $36 \pm 3.38$ | 0.029 | 0.39 |
|  | Imbalance in Stride Length between legs | $0.79 \pm 0.57$ | $1.01 \pm 0.63$ | 0.019 | $-0.43$ |
| Arm | Variation L Swing Velocity (°) | $41.28 \pm 23.52$ | $47.79 \pm 28.94$ | 0.032 | $-0.36$ |
|  | Variation R Swing Velocity (°) | $36.86 \pm 17.02$ | $44 \pm 25.63$ | 0.017 | $-0.38$ |
| Turning | Variation Turns Angle (°) | $5.7 \pm 1.44$ | $5.02 \pm 1.2$ | 0.010 | 0.48 |

All *p*-values reported are Bonferroni adjusted values: ° = degrees; L = left; R = right side; APA = Anticipatory Postural Adjustment, Flex/Ext = Flexion/Extension.

Individuals who reported feeling more fatigued had longer anticipatory postural adjustment time during gait initiation, lower variation in maximum lumbar right left rotation, and lower average lumbar extension ROM, less variation in trunk flexion/extension ROM, less variation in both right and left arm swing velocity, less imbalances in gait speed and stride length between legs, larger average toe out angles, and larger toe off angles for both the right and left legs. There were no differences in neck movement for individuals who reported being high fatigue compared to those who reported being low fatigue.

### 3.3. Energy and Fatigue Combined

There were 33 (males = 13, females = 20) participants in the high-energy/high-fatigue group, 40 (males = 10, females = 30) participants in the high-energy/low-fatigue group, 30 (males = 14, females = 16) in the low-energy/high-fatigue group, and 23 (males = 10, females = 13) in the low-energy/low-fatigue group. There were no significant differences in sex, age, height, weight, or BMI between groups (Table 1).

#### 3.3.1. Feature Importance

There were only four features that were important for the most accurate models: variations in maximum lumbar rotation ROM on the right side, variation in turn angle, anticipatory postural adjustment first step ROM, and variation in left leg double leg support time as a percentage of gait cycle time. The rest of the features had negative feature importance (see Table 2)

### 3.3.2. Model Training

The top-performing model was the Gradient Booster Classifier model with a mean accuracy of 43.14% (95% CI = 0.382–0.481), a median accuracy of 46.15%, and a mean AUC ROC of 0.658. The most accurate model had a 76.92% accuracy rate (Table 4).

### 3.3.3. Differences in Gait Characteristics between Groups

Only significant comparisons are presented (Table 7). Differences in all gait parameters are presented in Supplementary Table S1.

**Table 7.** Post hoc significant differences between groups for energy and fatigue.

| Variable | High Energy/Low Fatigue | High Energy/High Fatigue | Sig. | *d* |
|---|---|---|---|---|
| | Means ± SD | Means ± SD | | |
| Variation R Arm Swing Velocity (°/s) | 50.5 ± 26.9 | 35.24 ± 14.73 | 0.004 | 0.70 |
| Variation L Heel Strike Angle (°) | 1.90 ± 0.48 | 1.61 ± 0.35 | 0.033 | 0.69 |
| Variation R Arm ROM (°) | 8.66 ± 4.25 | 6.77 ± 2.29 | 0.045 | 0.55 |
| | **Low Energy/Low Fatigue** | **Low Energy/High Fatigue** | | |
| Variation Max. Lumbar R Rot ROM (°) | 6.57 ± 3.74 | 4.15 ± 2.7 | 0.018 | 0.74 |
| Variation Max. Back L Rot ROM (°) | 6.44 ± 3.82 | 4.11 ± 2.72 | 0.032 | 0.70 |
| | **High Energy/High Fatigue** | **Low Energy/High Fatigue** | | |
| Imbalance DLS-GCT between legs (%) | 0.42 ± 0.41 | 0.73 ± 0.42 | 0.045 | −0.75 |
| | **High Energy/Low Fatigue** | **Low Energy/High Fatigue** | | |
| Variation R Arm Swing Velocity (°/s) | 50.5 ± 26.9 | 38.19 ± 18.78 | 0.022 | 0.53 |
| Mean R Arm ROM (°) | 44.72 ± 16.81 | 36.56 ± 14.23 | 0.048 | 0.52 |
| | **High Energy/Low Fatigue** | **Low Energy/Low Fatigue** | | |
| Variation R Toe Off Angle (°) | 1.61 ± 0.5 | 1.3 ± 0.29 | 0.049 | 0.76 |
| Imbalance Arm Swing velocity between arms (°/s) | 209.95 ± 65.67 | 164.94 ± 40.06 | 0.042 | 0.83 |
| | **High Energy/High Fatigue** | **Low Energy/Low Fatigue** | | |
| Variation Turns Angle (°) | 6.07 ± 1.43 | 4.88 ± 1.13 | 0.018 | 0.92 |

All *p*-values presented in this table are Bonferroni adjusted ° = degrees; L = left; Max. = Maximum; Min. = minimum; R = right; ROM = range of motion; Rot = Rotation.

***High Energy/High Fatigue* vs. *High Energy/Low Fatigue***

Post hoc between-group comparisons showed that the high-energy and high-fatigue group had significantly less variation in left leg heel strike angle, variation in right arm swing velocity, and less variation in right arm ROM compared to individuals who reported being high-energy and low-fatigue.

***High Energy/High Fatigue* vs. *Low Energy/High Fatigue***

Group comparisons found that individuals who reported being highly energetic and highly fatigued had significantly less imbalance between legs for double leg support time compared to individuals who were low energy and high fatigue.

***High Energy/High Fatigue* vs. *Low Energy/Low Fatigue***

Individuals who reported being highly energetic and highly fatigued had greater variation in turn angles compared to individuals who reported being low energy and low fatigue.

*High Energy/Low Fatigue* **vs.** *Low Energy/High Fatigue*

Post hoc between-group comparisons found that individuals who reported being high energy and low fatigue had significantly greater variation in right upper arm swing velocity and greater mean right upper arm ROM compared to individuals who were low energy and high fatigue.

*High Energy/Low Fatigue* **vs.** *Low Energy/Low Fatigue*

Group comparisons found that individuals who reported being high energy and low fatigue had significantly higher variations in right leg toe off angle and slower arm swing velocity compared to individuals who were low energy/low fatigue.

*Low Energy/High Fatigue* **vs.** *Low Energy/Low Fatigue*

Individuals reporting low energy/high fatigue had significantly lower variation in maximum lumbar bending to the right and rotation ROM to the left side compared to individuals who were low energy and low fatigue.

## 4. Discussion

To the best of the knowledge of the researchers, this is the first study to use IMU sensors and machine learning to identify feelings of energy and fatigue, when measured as two separate moods, using single-task walking gait. Since previous literature using sensor technology and machine learning to identify subjective fatigue had measured fatigue as ratings of perceived exertion [11–16], there is no comparable literature to the findings of this study. Literature examining the effect of the association between feelings of energy and fatigue, as two unique moods, on walking gait does suggest that these two moods [31,32,35] influence unique aspects of gait, and the findings of this study support those works. These findings add three unique aspects to the literature: (1) when using machine learning to identify feelings of energy and fatigue, signals from IMU sensors during walking gait are unique to each mood; (2) machine learning models are able to predict these moods fairly accurately; and (3) there are advantages to identifying both feelings of energy and fatigue as two distinct unipolar moods in walking gait, as they influence ambulation in their own unique ways.

### 4.1. Comparing Most Important Features

When comparing the important features between the two models (low vs. high energy and low vs. high fatigue), the findings from this study suggest that all but one of the most important features differed between the models examining feelings of energy and fatigue. The only feature that was similar for both moods was anticipatory postural adjustment time for gait initiation, although the ranking of the feature was 11 for energy and 12 for fatigue, and the relative importance was ~2.6× greater for energy than it was for fatigue. The other 11 features were distinct, with both moods requiring the examination of a combination of variation in inter- and intra-limb movement, variation in midline movement and the mean values of certain features.

Additionally, adding weight as a feature to future models that seek to identify feelings of energy through walking gait would be important. These findings suggest that future researchers should make it a priority to examine inter-limb and intra-limb variations, as these variations may differ across moods, potentially impacting gait in a distinct manner. These results are further supported by the top features of the regressor models. These features also suggest that when identifying feelings of energy and fatigue on a spectrum, as is the case in this study (scores range from 0 to 20), the regressor models need to capture different signals for each mood, suggesting that feelings of energy and fatigue may influence different aspects of gait. Although we are unaware of comparable literature to this study, our findings do support previous works [31,32,35] that took a more traditional statistical approach to understanding how these two moods influence gait.

Feature importance findings for when the models were split into distinct classes of energy and fatigue (i.e., high energy/high fatigue, low energy/high fatigue) had only four features that had positive relative importance. The rest of the features had negative relative

importance, suggesting that these features made the models less accurate and that leaving these features out would be best. This may be explained by the relatively small sample of participants in each category (range 23–40), thus making it challenging to identify more gait features to classify these individuals. Taken together, the important feature results support the need to understand the unique influences of energy and fatigue as two distinct unipolar moods [22,24,26] on walking gait [31,32,35]. These results add to the literature by providing evidence that energy and fatigue influence different aspects of gait and capturing variations in inter- and intra-limb movements in addition to trunk movement and turning is of importance to researchers seeking to identify these moods using human gait.

### 4.2. Model Accuracy

The most accurate models for both feelings of energy and fatigue were Gradient Boosting Classifier models. While both reported similar mean accuracy (74.34% for energy vs. 74.23% for fatigue), the lower confidence interval for feelings of energy suggests that the identification of individuals who were low energy versus individuals who were high energy may be more accurate. Additionally, the best classifiers for energy had 100% accuracy, suggesting that if the model gets "lucky", and the "right" training and test dataset is used, feelings of energy may be identified very accurately using walking gait. While identification of feelings of fatigue had a similar mean accuracy, the higher confidence intervals, and the lower accuracy of the best and worst models suggests that other features may need to be captured to create more accurate models to identify individuals who are low fatigue versus those who are high fatigue. The mean AUC ROC was >0.80 for both models, suggesting that the models have an 80% + chance of distinguishing between the two classes created for both moods. These findings are not comparable to other literature; however, literature that has tried to identify other subjective moods, such as feelings of anxiety and depression, using a similar protocol has reported similar accuracy rates [39,40].

While the accuracy of measuring the two moods separately was in line with literature from studies examining other subjective mood states using walking gait [39,40], when classifying individuals using both moods simultaneously, the accuracy rates decline significantly (mean = 43.14%), and the AUC ROC shrinks (0.658) as well. The primary takeaways from these models are that: (1) there are individuals who may report being high energy and high fatigue ($n$ = 33), and those who can be both low energy and low fatigue ($n$ = 23), further supporting previously published findings [6–10]; (2) the number of individuals in each category may have been too small to create accurate machine learning models; (3) the AUC ROC for the best model (0.796) suggests that larger sample sizes may have yielded different results.

When examining the accuracy of the regressor models, both Energy and Fatigue models had very low mean MAE (mean MAE Energy = 0.005, mean MAE Fatigue = 0.007), both using K-fold cross-validation and bootstrapping techniques. When examining the coefficient of determination of the models ($R^2$), the mean $R^2$ for the models that used the bootstrap was high (mean $R^2$ Energy = 0.884, mean $R^2$ Fatigue = 0.886); however, when using K-fold cross-validation, the mean $R^2$ was reduced to 0.310 and 0.349 for Energy and Fatigue, respectively. A stratified K-fold with four strata was also used, but the $R^2$ was not very different. Taken together, the low MAE suggests that gait may be a good predictor of fatigue and energy [54]; however, the low $R^2$ in the K-fold models may be explained by the fact that there was a large variation in the data (scores ranging from 0 to 20), and a small sample size ($n$ = 126), and thus the K-fold cross-validation reduced the $R^2$. However, when examining bootstrapped models, the higher $R^2$ suggest that gait might help us identify feelings of energy and fatigue in larger samples, where K-fold cross-validation would result in higher values of $R^2$. The bootstrapped findings may have overfit the models; however, with small sample sizes, these models are the most appropriate analyses [51]. The results from both sets of models provide useful evidence for future researchers interested in identifying feelings of energy and fatigue using walking gait.

Together, the results of the classifier and regressor models support the use of walking gait to identify feelings of energy and fatigue using machine learning models.

### 4.3. Comparing Gait Characteristics between Groups

Comparing gait characteristics for the various groups provides evidence that energy and fatigue influence moods in unique ways. Visualizations of the various group gait parameters presented below can be found at https://gaitsim.dmanserver.com/EnergyFatigueMatrix/.

### 4.3.1. High Energy vs. Low Energy

When comparing low- and high-energy individuals, we found that low energy individuals had decreased variation in intra-limb movement; however, they had increased variation in inter-limb movement. These findings are in line with previous literature finding greater variation in inter-limb movement associated with decreased scores on objective [55] and subjective [35] indices of energy. Interestingly, these inter-limb variations and the mismatch between upper- and lower-extremity movement are patterns similar to literature examining the association between anxiety and gait [56,57], a mood associated with feelings of low energy [58]. Previous literature supports the association between arm swing ROM and feelings of energy [31,35]. Taken together, our findings suggest that individuals with low energy may be trying to counter-balance inter-limb variations in movement by reducing intra-limb variations to reduce error in walking gait and maintain their line of gravity within their base of support. Gait patterns in individuals with low energy are similar to those seen in individuals with increased risk of tripping [59].

### 4.3.2. High Fatigue vs. Low Fatigue

Individuals who reported high fatigue took longer to initiate gait, and exhibited gait patterns most commonly seen in individuals who are at low risk for falls [60,61]. These gait patterns included less imbalance in gait speed and stride length between legs, less imbalance in inter-arm swing velocity, and larger toe out angles, suggesting that these individuals were walking more symmetrically while increasing their base of support by rotating their foot outwards. Furthermore, less lumbar extension during gait suggests that these individuals may have been leaning forward, which may also explain the decreased variation in maximal lumbar rotation. These findings suggest a gait pattern similar to individuals who report being depressed [62,63], a mood associated with high feelings of fatigue [64]. It is also interesting that these individuals had greater variations in their turns, as this suggests that their symmetrical gait patterns during ambulation were making it harder for them to maintain turn angles, which is in line with previous findings [35]. Taken together, these findings suggest that individuals who report feeling more fatigued adopt ambulation strategies that reduce fall risks, with decreased lumbar extension [65], decreased gait variability, and increased base of support during ambulation [66].

### 4.3.3. Comparison between Classes

There are several differences between individuals who classify into the various classes. For example, individuals who were high energy and high fatigue tended to have more symmetrical movement in left leg heel strike angle, and also in right arm movement, compared to those who were high energy and low fatigue, suggesting that feelings of fatigue may impact turning (as individuals in this study were asked to turn left). These findings are similar to what has been reported when comparing the low- and high-fatigue groups and in a previous study identifying the interaction between walking gait and fatigue [35]. Similar findings can also be seen when comparing the low-energy individuals; the high-fatigue group had less variation in movements associated with torso stabilization when turning left compared to the low-fatigue group suggesting that high-fatigue individuals have trouble turning.

When comparing the high-fatigue groups, individuals who were low energy had greater variations in double leg support time compared to those who were highly energetic,

suggesting that individuals who are highly fatigued but also low energy have challenges regulating gait speed compared to those who are also high fatigue, but have high energy; patterns which are similar to previous findings [35,55]. However, when examining individuals who were low fatigue, those who also reported high energy had slower arm swing velocity and greater variations in toe off angle on the right foot. These minimal findings suggest that individuals who are high energy and also low fatigue may also be using strategies to reduce gait speed by decreasing arm swing velocity [35,55].

When comparing groups where feelings of energy and fatigue are in the expected directions (high energy/low fatigue vs. low energy/high fatigue), these findings report that individuals who were high energy/low fatigue had greater variations in their right arm movement during walking gait. Based on the fact that individuals in our study were asked to turn left during the two-minute walk, these results suggest that individuals who were high energy/low fatigue had more automatic gait and spent time correcting during their turns. Interestingly, individuals who were high energy and high fatigue had significantly greater variations in turn angles compared to those who were low energy and low fatigue, suggesting that a combination of the two moods influences turning.

*4.4. Limitations*

This study is not without limitations. The primary limitation of this study is the cross-sectional nature of the study. That is, the performance of an intervention to modify changes in feelings of energy and fatigue may have provided significant insight into the effects of these two moods on gait. However, due to inter-individual differences in how various interventions influence these two moods simultaneously [21,28,67,68], it may have been challenging to perform an exploratory study using an interventional design. Additionally, feelings of energy and fatigue are felt on a spectrum, and performing cross-validation to control such large variations in data requires large sample sizes, which impacted the $R^2$ of the K-fold cross-validated models in this study. Another potential limitation of this study is that the researchers did not measure feelings of energy and fatigue between the mCTSIB and the two-minute walk. However, there is no current literature available that suggests that two minutes of quiet balance influences feelings of energy and fatigue. Another limitation to the current study is that there were very few individuals in the groups when participants were categorized into groups based on both feelings of energy and fatigue. Due to the large number of variables in the analysis and the small group size, many of the comparisons were under-powered, thus resulting in many variables not being significant. This limited the interpretation of the gait differences between groups when classifying individuals based on feelings of both energy and fatigue simultaneously.

*4.5. Implications*

The results of this study have a significant impact on the literature in various fields. For example, identifying the intensity of feelings of energy and fatigue may be feasible using IMU sensors through walking gait. Additionally, researchers interested in studying psychological fatigue and its impact on walking gait should seek to differentiate between feelings of energy and fatigue, as the two moods seem to impact walking gait in unique ways. Based on the findings of this study, researchers interested in assessing the impact of psychological fatigue on fall risks in older adults would want to examine the impact of low energy on walking gait, as these findings suggest that low energy may result in increased errors in stride-to-stride variability during ambulation, making individuals more susceptible to trips and falls [60,61]. Interestingly, feelings of low energy were also associated with decreased balance control in a study of older adults, but not increased feelings of fatigue [30]. Conversely, individuals reporting increased feelings of fatigue tend to have gait patterns consistent with individuals who are at low risks for falls [60,61], suggesting that feelings of fatigue may serve to act protectively. Further researchers interested in identifying psychological energy and fatigue should seek to use computer

vision as the findings of this current study suggest that using single sensors to identify these mood states may not be feasible.

## 5. Conclusions

To the best of the authors' knowledge, this is the first study to quantify gait characteristics associated with self-reported psychological feelings of energy and fatigue using machine learning and IMU sensors. In a cross-sectional study of 126 participants, the findings support previously published evidence that energy and fatigue may influence distinct aspects of gait [32,35], as the important feature selection of the machine learning models were distinct for the two moods. Furthermore, the results of this study indicate that machine learning models to identify psychological fatigue and energy through walking gait using IMU sensors were fairly accurate in this sample. These results provide gait researchers, clinicians, and computer scientists interested in studying psychological fatigue evidence that feelings of energy and fatigue must be measured as two distinct moods. These results also support the need to measure movements in both the upper and lower extremities, the trunk, the neck, and the lumbar spine. Future research should seek to influence feelings of energy or fatigue and determine how these changes (both increases and decreases) uniquely influence single-task walking gait.

**Supplementary Materials:** The following supporting information can be downloaded at: https://www.mdpi.com/article/10.3390/app12063083/s1, Table S1: All Gait Features.

**Author Contributions:** Conceptualization: A.T., A.M.K., M.A.E., A.B.; Methodology: A.B.; Formal analysis: A.T., A.M.K., M.A.E., S.C., A.B.; Investigation: A.B.; Data curation: A.T., A.M.K., M.A.E., S.C., A.B.; writing original draft preparation: A.T., A.M.K., M.A.E., S.C., R.S.K., D.B., A.B.; Visualization: D.B., A.T., A.M.K., M.A.E.; Supervision: A.B.; Writing review and editing: R.S.K., S.C., A.B. All authors have read and agreed to the published version of the manuscript.

**Funding:** This research received no external funding. The authors A.T., A.M.K. and M.A.E. are funded by USAID in the lab of author A.B.

**Institutional Review Board Statement:** This study was conducted according to the guidelines of the Declaration of Helsinki, and approved by the Institutional Review Board of Clarkson University (approval #18.39.1).

**Informed Consent Statement:** Informed consent was obtained from all subjects involved in the study.

**Data Availability Statement:** Data are available upon request due to Institutional Review Board (IRB) restrictions.

**Acknowledgments:** The authors would like to recognize Guilia Mahoney, Maggie Stark, Chelsea Yager, Rebecca Martin, Seema Teymouri, Christina Vogel-Rosbrook, and Phylicia Taladay for their help with data collection. The authors would also like to recognize Chelsea Yager and Rebecca Martin with their help in submitting the IRB.

**Conflicts of Interest:** The authors have no conflict of interest to report.

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
