# Peer review of "Using Machine Learning to Identify Feelings of Energy and Fatigue in Single-Task Walking Gait: An Exploratory Study"

_applsci, doi:10.3390/app12063083_

Round 1

Reviewer 1 Report

In this paper, the authors have tested a variety of algorithms to determine if mood and fatigue can be classified based on gait characteristics. Gait characteristics are also compared between mood and fatigue levels. The authors found that different variables were selected to classify mood compared to fatigue and conclude that these affect gait in different ways and should be considered separately. I think the premise of the study is interesting; however, there are details that need to be clarified and I have some concerns that very small differences detected between groups are overinterpreted.

It would be helpful to indicate in the methods section that you are considering 168 different variables and give a brief description of the type of variables that are included. It would also be useful to explain some of the variables that are discussed later when the meaning of the measure is not obvious (e.g., leg circumduction or anticipatory postural adjustment).

On page 5, equations are given for “variance”; however, it appears you are actually computing the standard deviation (i.e., square root of variance). The equations here are also formatted inconsistently and imprecisely.

Does “gait speed right leg” refer to the speed of the body centre of mass while the right leg is on the ground or something like that, or does it refer to the speed of the leg itself? In either case, I’m not sure how the sum of the right and left leg speed would give you an overall speed, as opposed to the mean of the two. Perhaps some of the terms here can be clarified?

Is the equation for variance of gait parameters between limbs a standard measure? This does not seem to be a variance. It also seems like there may be an error in this equation, if I am interpreting the variables correctly. It looks like it is saying to normalize the difference between right and left by the sum of right and left at each frame, them sum these up across all frames of data and multiply by 100, which doesn’t make sense to me.

Why was it necessary to fill in missing values as opposed to just not including them in the calculation of the mean and variance?

Why was the data winsorized? Were there extreme values that were affecting the mean and standard deviation? And did you investigate these to determine why those values might have been so extreme (i.e., is there an error, or is it real data)? The decision to arbitrarily remove all “outliers” should be better justified if it is included.

What is the outcome measure of the regression model? Is it a continuous value between 0 and 20, or is it an integer value?

On page 6, it is stated that the “model was run” 10,000 times. Do you mean that the process of splitting the data into test and training sets then training/testing the model was repeated this many times? If so, I don’t believe this would be considered a 10-fold cross-validation, as that involves repeating this process 10 times.

Can you clarify the statistical analysis used? MANCOVA usually stands for Multivariate Analysis of Covariance and mixed model usually means you are including both with and between subject factors, so I’m a bit confused about what was done. It’s also not clear to me what then numbers in “2x2” and “4x2” are referring to.

Were all 168 outcome measures included in the MANCOVA and then corrected for in your post hoc analysis?

What do you mean by “regressor models are presented both with and without K-fold cross-validation”? Do you mean you are also reporting the result training data only when the entire data set is used for training? If this is the case, then a possible explanation for the much better results for the models without k-fold validation could simply be overfitting.

The visualization is neat. There appears to be something odd happening with the foot motion though. Is this the foot movement measured by the sensors used?

Was feature importance determined based on the ability of variables to classify groups or something else? It seems odd that the most important features did not always differ between groups, but other variables did.

Many of the differences identified and extensively discussed are extremely small (e.g., 0.3s or 0.007 degrees). Do these really have any practical relevance?

It is stated in 4.3.1 that those with low energy have faster gait speed. However, the right gait speed is greater in those with low energy. Although, as stated above, this difference is so small that I’m doubtful that this (or the other differences listed) have any practical meaning.

Most of the tables are cut off so I can’t see large portions of them. Maybe it’s because of this, but I also don’t understand the explanation under the tables. There seems to be one importance measure reported, but there are two methods of calculating importance listed.

Please spell out all acronyms the first time they are used (e.g. in Sub-Objective 2).

There are minor grammatical errors throughout that should be corrected and the formatting of comparisons in the results section is inconsistent and sometimes missing units.

Reviewer 2 Report

Introduce a multybody models to describe postural sway. Example

One work by Pascolo and Saccavini concerning erect stance, you can se

e it on web or pub med

Reviewer 3 Report

The aim of this study was to use machine learning to identify feelings of energy and fatigue using single-task walking gait. The article reports an innovative and interesting work. Indeed, increasing attention is being paid to  machine learning. 
Innovation and interest in the study are unfortunately overshadowed by the confusion with which the results and discussion are presented. From the introduction, the data in this paper seem to be useful for preventing falls in the elderly. However, this study is carried out on a young population. It would be advisable to reformulate the introduction, summarising it so as to make it more usable for the busy reader. In particular, the aims of the study should emerge more from the introduction in order to make clear why this interesting analysis was carried out. 
The manuscript is written in good English, however there are some sections that need to be further elaborated. The aims of the study could be further simplified to make them clearer.
The materials and methods section is precisely described so that the procedure followed is repeatable. Statistical analysis is well conducted and appropriate.
 "A combination of Chi-squares and student T-tests were used to identify differences in sex distribution, age, height, weight and Body Mass Index (BMI) between each group. " I suggest that this sentence should be further elaborated.
I suggest simplifying the results, which are very numerous and could be shown in tables, making the text easier to understand. I would also avoid repetitions between text and tables.
I also suggest implementing the discussion with more references to the literature.
Overall, I suggest that the strength of this article and its innovation should be highlighted more in the text, avoiding excessively long paragraphs that lose sight of the objectives of the study. I suggest revising the text as a whole, including the structure into which it is divided, to make it more accessible. 

Round 2

Reviewer 3 Report

Suggestions were accepted.